# Association Between Common Variants in the *LAG3*/*CD4* Genes and Risk for Essential Tremor

**DOI:** 10.3390/ijms252413403

**Published:** 2024-12-13

**Authors:** José A. G. Agúndez, Yolanda Macías, Hortensia Alonso-Navarro, Elena García-Martín, Ignacio Álvarez, Pau Pastor, Julián Benito-León, Tomás López-Alburquerque, Félix Javier Jiménez-Jiménez

**Affiliations:** 1University Institute of Molecular Pathology Biomarkers, Universidad de Extremadura, 10071 Cáceres, Spain; jagundez@unex.es (J.A.G.A.); yolandamg@unex.es (Y.M.); elenag@unex.es (E.G.-M.); 2Section of Neurology, Hospital Universitario del Sureste, 28500 Arganda del Rey, Spain; hortalon@yahoo.es; 3Fundació per la Recerça Biomèdica i Social Mútua de Terrassa, 08221 Terrassa, Spain; ignacioalvafer@gmail.com (I.Á.); pastorpau@gmail.com (P.P.); 4Movement Disorders Unit, Department of Neurology, Hospital Universitari Mutua de Terrassa, 08221 Terrassa, Spain; 5Hospital Universitari Germans Trias y Pujol, 08916 Badalona, Spain; 6Service of Neurology, Department of Medicine, Hospital Doce de Octubre, Universidad Complutense, 28040 Madrid, Spain; jbenitol67@gmail.com; 7CIBERNED, Centro de Investigación Biomédica en Red de Enfermedades Neurodegenerativas, Instituto de Salud Carlos III, 28222 Madrid, Spain; 8Department of Neurology, Hospital Universitario de Salamanca, 37007 Salamanca, Spain; albur@usal.es; 9Department of Medicine-Neurology, Hospital “Príncipe de Asturias”, Universidad de Alcalá, 28801 Alcalá de Henares, Spain

**Keywords:** essential tremor, genetics, gene variants: LAG gene, CD4 gene, risk factors

## Abstract

Many clinical, neuroimaging, neuropathological, epidemiological, and genetic data suggest a relationship between essential tremor (ET) and Parkinson’s disease (PD). Several hypothesis-based gene association studies attempted to find a genetic association between these diseases. Recent case–control association studies in Chinese and Spanish populations showed a marginal association between the *CD4* rs1922452 and *CD4* rs951818 single nucleotide variants (SNVs) and the risk of PD. The proteins encoded by the *CD4* and *LAG3* genes have an important role in modulating inflammatory responses, and some recent data associated inflammatory markers to ET. This study investigates a possible association between the most common SNVs in the *LAG3/CD4* genes and the risk of ET in the Spanish Caucasian population. We genotyped 267 patients diagnosed with familial ET and 270 age- and sex-matched controls using specific TaqMan assays for *CD4* rs1922452, *CD4* rs951818, and *LAG3* rs870849 variants. We found a decreased risk for ET in carriers of the *LAG3* rs870849 C/C genotype and the *LAG3* rs870849C allelic variant exclusively in men. The mean age of onset of ET was not related to any of the variants studied. These data suggest no association of the gene variants studied with the overall risk for ET, except for a slight decrease in risk in male ET patients carrying the variant *LAG3* rs870849C. However, such an association lost significance after correcting for multiple comparisons.

## 1. Introduction

Essential tremor (ET) is a movement disorder of high prevalence, especially in old age [1,2,3,4,5,6]. According to a recent meta-analysis, the prevalence of ET in the general population was 0.32% (0.36% in males and 0.28% in females), rising to 4.16% to 5.85% in individuals over 60 years [7]. ET is characterized from the clinical point of view by the presence of postural tremor affecting fundamentally upper limbs [1,2,3,4,5]. However, a variable percentage of patients show tremor in other body locations, and many of them are associated with non-motor symptoms or co-morbidities [1,2,3,4,5]. The important role of genetic factors in the etiopathogenesis of ET is supported by the following [8,9,10,11,12,13]: (a) the higher concordance rates of ET for monozygotic twins compared with dizygotic ones, (b) the high positivity of family history of tremor, and (c) the presence of genetic anticipation. In addition, the contribution of environmental factors, at least in non-familial or sporadic cases of ET has also been suggested [14,15,16].

In the last three decades, numerous studies have been published trying to establish the possible contribution of genetic factors in the etiopathogenesis of ET. Thus, genetic linkage studies have identified four susceptibility loci associated with familial ET, although without identifying the responsible gene(s) [8,9,10,11,12]. In addition, several variants in the *LINGO1*, *SLC1A2*, *STK32B*, *PPARGC1A*, and *CTNNA3* genes have shown an association with the risk for ET in Genome-wide Association Studies (GWAS). However, these associations were not confirmed in replication studies [8,9,10,11,12]. The association of ET with variants in the *PTGFRN* (rs1127215), *PPARGC1A* (rs17590046), *MIR924HG* (rs1945016), *LOC105379011* (rs28562175), and *LINC00323* (rs9980363) genes have been found in a recent GWAS and meta-analysis involving 7177 ET patients and 475,877 controls [12]. The association between familial ET (although limited to a few families) and many genes has been described. These associations involve genes, such as *FUS*, *HTRA2*, *TENM4*, *NOS3*, *KCNS2*, *SLIT3*, *CCDC183*, *SCN11A*, *NOTCH2NLC*, *USP46*, *CACNA1G*, *SORT1*, *HAPLN4*, *MMP10*, and *GPR151*, have been described by exome sequencing in families and case series of patients with ET [11]. Finally, hypothesis-driven case–control studies described an association of variants in several candidate genes with the risk of ET, including *CYP2C19* [17], *CYP2C9/CYP2C8* [18], *RIT2* [19], *IL1B* [20], *VDR* [21,22] and *GC* [22]. Also, an increased risk for carriers of the *ALAD* rs1800435 variants in interaction with a variant in the *HMOX2* gene [23] and with serum levels of lead [24] has been described. However, most of these associations have not been replicated.

Lymphocyte activation gene 3 protein (LAG3) is expressed by both activated and exhausted CD4+ and CD8+ T cells, regulatory T cells, and microglia. This protein acts by delivering inhibitory signals that regulate immune cell homeostasis, T cell activation, proliferation, cytokine production, cytolytic activity, and other functions related to inflammatory responses [25,26], and is encoded by the *LAG3* or *CD223* gene (chromosome 12p13.31; MIM 153337, gene ID 3902), which is closely related to the *CD4 molecule* gene, (*CD4*; MIM 186940; gene ID 920), both at the gene and protein levels [27]. *CD4* gene encodes the CD4 membrane glycoprotein of T lymphocytes (an important mediator of inflammatory and immune responses). CD4 protein is expressed in the brain of adult rats, including in neurons from the cerebellar cortex (both in granule cells and in Purkinje cells) and thalamus (implicated in the generation of tremor). Also, albeit to a lesser extent, it is expressed in the striatum and substantia nigra compacta [28]. In the human brain, CD4 is highly expressed in the thalamus and basal ganglia and has some degree of expression in the cerebellum as well [29]. The presence or absence of LAG3 protein in the brain is under debate, with some publications indicating its presence in certain brain regions and others finding a lack of evidence of its presence in neuronal cell lines [30].

Many sets of clinical, neuroimaging, neuropathological, epidemiological, and genetic data suggest a relationship between ET and PD. This relationship is supported by many epidemiological, genetic, clinical, neuropathological, and neuroimaging data [31,32]. In addition, several hypothesis-driven case–control association studies of genes related to monogenic familial PD have been carried out to look for possible genomic markers for ET [8,10,11]. In this regard, recent case–control association studies in Chinese [33] and Caucasian Spanish populations [34] showed a marginal association between *CD4* rs1922452 and *CD4* rs951818 single nucleotide variants (SNVs) and the risk for Parkinson’s disease. Moreover, some recent preliminary data suggest a possible relation between inflammatory factors and ET [20,35,36,37], with LAG3 and CD4 proteins implicated in the inflammatory responses. The current work investigates a possible association between the most common SNVs in the *LAG3*/*CD4* genes and the risk for ET in the Caucasian Spanish population.

## 2. Results

No statistically significant departures from Hardy–Weinberg’s equilibrium were observed for the genotypes *CD4* rs1922452, *CD4* rs951818, and *LAG3* rs870849, both in ET patients and in control groups. The Chi-square values for the Hardy–Weinberg equilibrium were as follows: rs1922452 A/G, 0.023 in patients, and 0.003 in controls; rs951818 A/C, 0.281 in patients, and 0.126 in controls; rs870849 C/T, 0.001 in patients, and 0.144 in controls. The frequencies of the genotypes and allelic variants *CD4* rs1922452, *CD4* rs951818, and *LAG3* rs870849, did not differ significantly between ET patients and controls considering the whole series (Table 1) and female gender (Table 2). However, the frequencies of the *LAG3* rs870849 C/C genotype and *LAG3* rs870849C allelic variant were significantly lower in men with ET compared with men in the control group. The statistical significance remained after correction for multiple comparisons (Table 2). No statistically significant departures from the Hardy–Weinberg equilibrium were present for any of the genotypes studied after classifying participants according to sex (Table 2) or to tremor localization. The Chi-square values for Hardy–Weinberg’s equilibrium were as follows: rs1922452 A/G, 0.148 in women with ET, 0.005 in control women, 0.020 in men with ET, and 0.000 in control men; rs951818 A/C, 0.919 in women with ET, 0.046 in control women, 0.048 in men with ET, and 0.086 in control men; rs870849 C/T, 0.130 in women with ET, 0.046 in control women, 0.041 in men with ET, and 0.474 in control men. Regarding the putative linkage disequilibrium between the variants analyzed, the D’ statistics in the study group were as follows: rs1922452 vs. rs951818 = 0.9705; rs1922452 vs. rs870849 = 0.0879; and rs951818 vs. rs870849 = 0.0387. The R2 statistics were as follows: rs1922452 vs. rs951818 = 0.9232; rs1922452 vs. rs870849 = −0.054; and rs951818 vs. rs870849 = −0.025. These findings indicate that the SNPs rs1922452 and rs951818 are almost at complete linkage disequilibrium, whereas rs870849 is not in disequilibrium either with rs1922452 or rs951818. These findings are consistent with data corresponding to the Iberian population in Spain corresponding to the GRCh37 genome build (https://ldlink.nih.gov, accessed on 25 November 2024).

The mean ± SD age at onset of tremor did not differ significantly between the three different genotypes of each of the three variants studied (Table 3).

## 3. Discussion

It has been proposed that neuroinflammation is particularly important in the pathophysiology of neurodegeneration, and results from pathological studies have shown the presence of neurodegeneration in the brains of patients diagnosed with ET [38]. Despite reports of the presence of neuroinflammation in neurodegenerative diseases such as Parkinson’s [39,40] and Alzheimer’s diseases [41], data regarding the possible role of neuroinflammation in ET are scarce and inconclusive. A recent study involving 90 ET patients and 90 healthy controls showed significantly lower serum Tumor Necrosis Factor-α (TNF-α), and higher serum interleukins 8 and 10 (IL-8 and IL-10) levels in patients with ET. Such a study showed also a correlation between serum IL-10 levels and the severity of tremor and higher serum IL-6 levels in patients with ET and cognitive impairment [35]. In contrast, Tak & Sengül [36] reported no significant differences in two markers for inflammation, neutrophil-to-lymphocyte ratio (NLR) and platelet-to-lymphocyte ratio (PLR), between 67 ET patients and 40 healthy controls. Finally, Chen et al. [37] showed no significant differences between 105 patients with ET and 148 healthy controls in the serum levels of anti-*Saccharomyces cerevisiae* antibody (ASCA, a marker of chronic gastrointestinal inflammation). Other authors described an association of IL1B polymorphisms, related to pro-inflammatory processes, with the risk for ET [20].

The main result of the present case–control association study was the lack of a major association between the *CD4* rs1922452, *CD4* rs951818, and *LAG3* rs870849 variants and the risk for ET. Compared with male controls, men with ET showed a significant decrease in the frequency of *LAG3* rs870849CC genotype and *LAG3* rs870849C allele. However, when applying tests for multiple comparisons, the significance of this finding was marginal (Pc = 0.06), and, therefore, should be considered negative. In addition, these data might be considered preliminary and need to be replicated in future studies.

Figure 1 shows Forest Plots including OR and 95% CI for all variants in overall ET patients as well as patients stratified by sex (Figure 1, panel A). Also, a comparative analysis of each variant among patients with ET (this study) and patients with Parkinson’s disease [34] is shown in Figure 1, panel B. The comparison of the results obtained in ET with those obtained in PD patients indicates that the association obtained with PD is not evident in ET patients.

Several variants in the *LAG3* and *CD4* genes have been associated with the risk for multiple sclerosis (rs1922452 A > G) [42] and Parkinson’s disease [33,34]. Some of these variants were related to disease progression and mortality of sepsis (rs951818 C > A) [43] and the severity of primary immune thrombocytopenia (rs870849 T > C) [44].

The current study has limitations such as the low sample size. Moreover, despite the controls being sex-matched with cases, they were not matched by age. It should be stated, however, that the mean ± SD age of controls was close to that of the age at onset of tremor in ET patients, as shown in Figure 2.

Therefore, we must consider the possibility that some individuals in the control group could develop ET in the future. Considering the prevalence of ET in Spain [45], it is unlikely that the percentage of subjects with risk genotypes who would develop this disease in the time lapse between the average ages of patients with ET and controls would be high enough to modify the results of the study. Considering these limitations of the study, our results suggest a lack of association between the most common variants of the LAG3/CD4 genes and the risk of developing ET. The only exception is the slight, nearly significant decrease in risk in male patients carrying the *LAG3* rs870849C variant. Future replication studies with larger sample sizes trying to explore gender-specific effects are warranted.

## 4. Patients and Methods

### 4.1. Patients and Controls

We studied the genotype and allelic variants *CD4 rs1922452, CD4 rs951818*, and *LAG3 rs870849* variants, in 267 ET patients and 270 gender-matched controls, all of them of Caucasian Spanish origin. Standard diagnostic criteria for the diagnosis of definite ET [46] were used to select ET patients involved in the study (130 men and 137 women, mean age 66.4 ± 15.9 years, mean age at onset of tremor 49.6 ± 25.3 years), all of them with at least one first-degree relative also diagnosed with ET. In addition, inclusion in the study required the absence of a family history of other neurological diseases, including PD, and normal thyroid function studies. Patients with ET, many of whom had participated in other genetic case–control association studies published by our group [8,11,17,18,22,23], were recruited from the movement disorders units of several University Hospitals. Controls were recruited from the University of Extremadura. These were unrelated healthy subjects of Spanish European ancestry who did not have a personal or familial history of tremor, PD, or other movement disorders (130 men and 140 women, mean age 51.1 ± 14.5 years). Among ET patients, 75 had head tremor, 48 had voice tremor, 32 had tongue tremor, and 21 had chin tremor.

### 4.2. Ethical Aspects

The study protocol was approved by the Ethics Committees of the University Hospital “Príncipe de Asturias” (Alcalá de Henares, Madrid, Spain, 2004, no reference code number), and of the University Hospital “Infanta Cristina” (Badajoz, Spain, 2004, no reference code number). The study was performed following the principles of the Declaration of Helsinki, which is an obligate requisite for obtaining written and signed informed consent.

### 4.3. Genotyping of CD4 rs1922452, CD4 rs951818, and LAG3 rs870849 Variants

Genotyping studies were performed by obtaining genomic DNA from peripheral leukocytes of venous blood samples of patients diagnosed with ET and controls. The analysis was performed by using real-time PCR (Applied Biosystems 7500 qPCR thermocycler, Foster City, CA, USA) with specific TaqMan probes. All probes were purchased from Life Technologies (Alcobendas, Madrid, Spain). These probes were the following: (C__11914936_10) for rs1922452, which consists of an A to G substitution in chromosome 12-6896194 (GRCh37); (C___8921385_10) for rs951818, which consists of a C to A substitution in chromosome 12-6896055; and (C___9797874_10) for rs870849, which consists of a T to A substitution in chromosome 12-6777854. These SNVs were, respectively, one intronic and one non-coding transcript exonic SNV with high allele frequencies, and the only missense SNVs with an allele frequency over 0.01 in the population involved in this study.

### 4.4. Statistical Analysis

We used the SPSS 27.0 version for Windows (SPSS Inc., Chicago, IL, USA) to perform the statistical analysis, and the online program https://www.snpstats.net/start.htm (accessed on 24 November 2024) to explore the Hardy–Weinberg for the 3 variants assessed, both in ET patients and control groups. The Chi-square or Fisher’s exact tests, where appropriate, were used to calculate the intergroup comparison values (both for the whole series and each gender of ET patients and controls). The 95% confidence intervals (95% CI) [47], and the correction for adjustment for multiple analyses by using the False Discovery Rate (FDR) [48], were also calculated. This procedure is adequate for the control of type I error by determining the proportion of wrongly rejected null hypotheses amongst those that are rejected (instead of amongst all). This procedure is more sensible when there is a high number of comparisons as in this study: three genotypes x three groups (overall patients, men and women) x five groups according to tremor localization.

The statistical power (two-tailed association) for the *CD4* rs1922452, *CD4* rs951818, and *LAG3* rs870849 variant alleles, according to the sample size of this study, and using a genetic model that analyzed the minor allele frequencies with an odds ratio (OR) value = 1.5 (α = 0.05), was 90.99%, 91.06%, and 90.68%, respectively. For gender-specific subgroup analyses, the two-tailed statistical power was as follows: CD4 rs1922452—women 62.62%, men 59.14%; CD4 rs951818—women 65.96%, men 63.06%; LAG3 rs870849—women 65.96%, men 61.32%.

Finally, the Student’s *t*-test was used to compare the mean ± SD age at onset of tremor across the different genotypes of the SNVs studied.

## Figures and Tables

**Figure 1 ijms-25-13403-f001:**
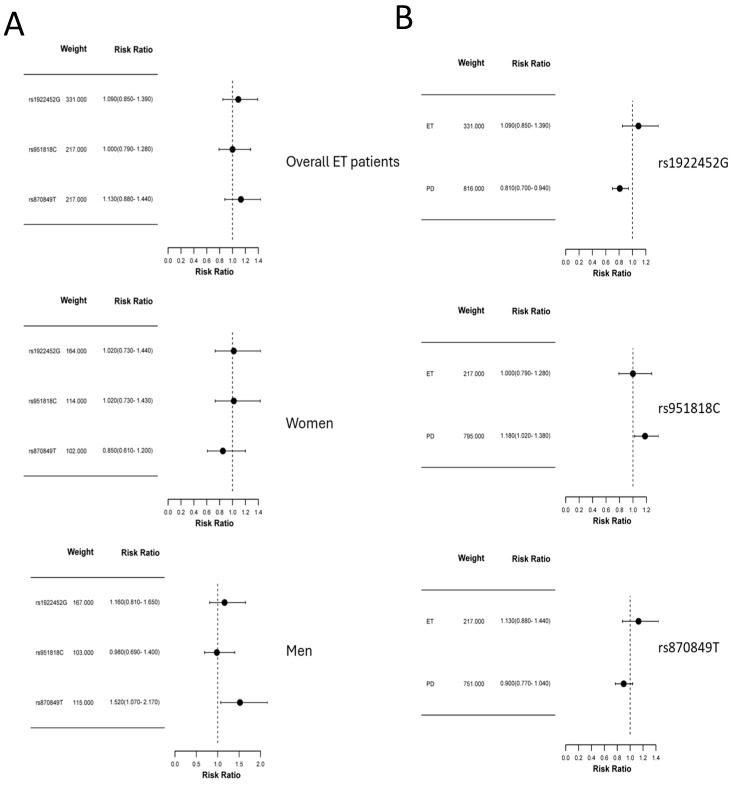
Forest plot of OR and 95% confidence intervals for all variants in overall ET patients as well as patients stratified by sex (**Panel A**) and comparative analysis of each variant among patients with ET (this study) and patients with Parkinson’s disease [34] (**Panel B**).

**Figure 2 ijms-25-13403-f002:**
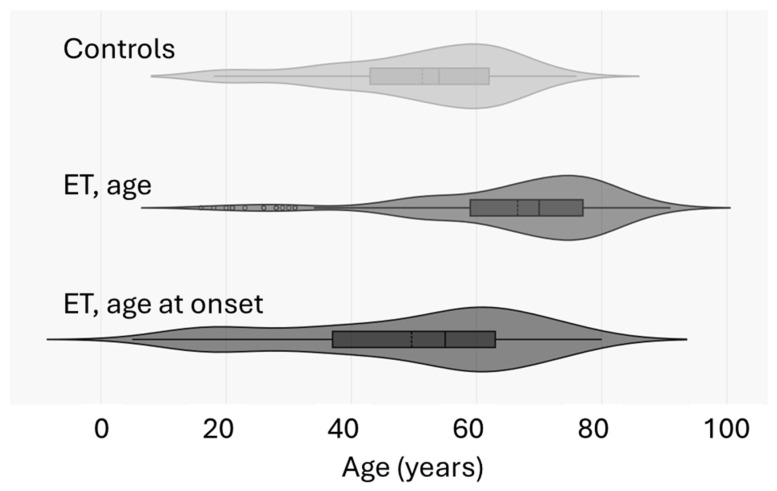
Violin plot of age, age at onset of TE patients, and age of controls.

**Table 1 ijms-25-13403-t001:** Genotypes and allelic variants of patients with essential tremor and healthy volunteers. The values in each cell represent the number (percentage; 95% confidence intervals). P: crude probability; Pc: probability after multiple comparisons.

Genotype	ET Patients (*n* = 267, 534 Alleles)	Controls (*n* = 270, 540 Alleles)	OR (95% CI); P; Pc
rs1922452 A/A	38 (14.2; 10.0–18.4)	43 (15.9; 11.6–20.3)	0.88 (0.55–1.41); 0.584; 0.951
rs1922452 A/G	127 (47.6; 41.6–53.6)	130 (48.1; 42.2–54.1)	0.98 (0.70–1.37); 0.893; 0.951
rs1922452 G/G	102 (38.2; 32.4–44.0)	97 (35.9; 30.2–41.6)	1.10 (0.78–1.57); 0.585; 0.951
rs951818 A/A	92 (34.5; 28.8–40.2)	94 (34.8; 29.1–40.5)	0.98 (0.69–1.41); 0.931; 0.951
rs951818 A/C	133 (49.8; 43.8–55.8)	133 (49.3; 43.3–55.2)	1.02 (0.73–1.43); 0.898; 0.951
rs951818 C/C	42 (15.7; 11.4–20.1)	43 (15.9; 11.6–20.3)	0.99 (0.62–1.57); 0.951; 0.951
rs870849 C/C	94 (35.2; 29.5–40.9)	106 (39.3; 33.4–45.1)	0.84 (0.59–1.19); 0.332; 0.951
rs870849 C/T	129 (48.3; 42.3–54.3)	124 (45.9; 40.0–51.9)	1.10 (0.78–1.54); 0.580; 0.951
rs870849 T/T	44 (16.5; 12.0–20.9)	40 (14.8; 10.6–19.1)	1.14 (0.71–1.81); 0.596; 0.951
**Allele**			
rs1922452 A	203 (38.0; 33.9–42.1)	216 (40.0; 35.9–44.1)	0.92 (0.72–1.18); 0.505; 0.756
rs1922452 G	331 (62.0; 57.9–66.1)	324 (60.0; 55.9–64.1)	1.09 (0.85–1.39); 0.505; 0.756
rs951818 A	317 (59.4; 55.2–63.5)	321 (59.4; 55.3–63.6)	1.00 (0.78–1.27); 0.978; 0.987
rs951818 C	217 (40.6; 36.5–44.8)	219 (40.6; 36.4–44.7)	1.00 (0.79–1.28); 0.978; 0.978
rs870849 C	317 (59.4; 55.2–63.5)	336 (62.2; 58.1–66.3)	0.89 (0.69–1.13); 0.337; 0.756
rs870849 T	217 (40.6; 36.5–44.8)	204 (37.8; 33.7–41.9)	1.13 (0.88–1.44); 0.337; 0.756

**Table 2 ijms-25-13403-t002:** Genotypes and allelic variants of patients with essential tremor and healthy volunteers stratified by sex. The values in each cell represent the number (percentage; 95% confidence intervals). P: crude probability; Pc: probability after multiple comparisons.

Genotype	ET Women (*n* = 137, 274 Alleles)	Control Women (*n* = 140, 280 Alleles)	OR (95% CI); P; Pc	ET Men (*n* = 130, 260 Alleles)	Control Men (*n* = 130, 260 Alleles)	OR (95% CI); P; Pc
rs1922452 A/A	21 (15.3; 9.3–21.4)	23 (16.4; 10.3–22.6)	0.92 (0.48–1.76); 0.803; 0.977	17 (13.1; 7.3–18.9)	20 (15.4; 9.2–21.6)	0.83 (0.41–1.66); 0.595; 0.865
rs1922452 A/G	68 (49.6; 41.3–58.0)	68 (48.6; 40.3–56.9)	1.04 (0.65–1.67); 0.860; 0.977	59 (45.4; 36.8–53.9)	62 (47.7; 39.1–56.3)	0.91 (0.56–1.48); 0.710; 0.865
rs1922452 G/G	48 (35.0; 27.0–43.0)	49 (35.0; 27.1–42.9)	1.00 (0.61–1.64); 0.995; 0.995	54 (41.5; 33.1–50.0)	48 (36.9; 28.6–45.2)	1.21 (0.74–2.00); 0.447; 0.865
rs951818 A/A	44 (32.1; 24.3–39.9)	48 (34.3; 26.4–42.1)	0.91 (0.55–1.50); 0.702; 0.967	48 (36.9; 28.6–45.2)	46 (35.4; 27.2–43.6)	1.07 (0.64–1.77); 0.797; 0.865
rs951818 A/C	72 (52.6; 44.2–60.9)	69 (49.3; 41.0–57.6)	1.14 (0.71–1.83); 0.587; 0.967	61 (46.9; 38.3–55.5)	64 (49.2; 40.6–57.8)	0.91 (0.56–1.48); 0.710; 0.865
rs951818 C/C	21 (15.3; 9.3–21.4)	23 (16.4; 10.3–22.6)	0.92 (0.48–1.75); 0.803; 0.967	21 (16.2; 9.8–22.5)	20 (15.4; 9.2–21.6)	1.06 (0.54–2.07); 0.865; 0.865
rs870849 C/C	53 (38.7; 30.5–46.8)	48 (34.3; 26.4–42.1)	1.21 (0.74–1.97); 0.448; 0.967	41 (31.5; 23.6–39.5)	58 (44.6; 36.1–53.2)	0.57 (0.35–0.95); 0.030; 0.270
rs870849 C/T	66 (48.2; 39.8–56.5)	69 (49.3; 41.0–57.6)	0.96 (0.60–1.53); 0.854; 0.967	63 (48.5; 39.9–57.1)	55 (42.3; 33.8–50.8)	1.29 (0.79–2.09); 0.320; 0.865
rs870849 T/T	18 (13.1; 7.5–18.8)	23 (16.4; 10.3–22.6)	0.77 (0.40–1.50); 0.442; 0.967	26 (20.0; 13.1–26.9)	17 (13.1; 7.3–18.9)	1.66 (0.85–3.24); 0.134; 0.603
**Allele**						
rs1922452 A	110 (40.1; 34.3–46.0)	114 (40.7; 35.0–46.5)	0.98 (0.70–1.37); 0.892; 0.898	93 (35.8; 29.9–41.6)	102 (39.2; 33.3–45.2)	0.86 (0.61–1.23); 0.415; 0.625
rs1922452 G	164 (59.9; 54.0–65.7)	166 (59.3; 53.5–65.0)	1.02 (0.73–1.44); 0.892; 0.898	167 (64.2; 58.4–70.1)	158 (60.8; 54.8–66.7)	1.16 (0.81–1.65); 0.415; 0.625
rs951818 A	160 (58.4; 52.6–64.2)	165 (58.9; 53.2–64.7)	0.98 (0.70–1.37); 0.898; 0.898	157 (60.4; 54.4–66.3)	156 (60.0; 54.0–66.0)	1.02 (0.72–1.44); 0.929; 0.929
rs951818 C	114 (41.6; 35.8–47.4)	115 (41.1; 35.3–46.8)	1.02 (0.73–1.43); 0.898; 0.898	103 (39.6; 33.7–45.6)	104 (40.0; 34.0–46.0)	0.98 (0.69–1.40); 0.929; 0.929
rs870849 C	172 (62.8; 57.0–68.5)	165 (58.9; 53.2–64.7)	1.17 (0.84–1.65); 0.354; 0.898	145 (55.8; 49.7–61.8)	171 (65.8; 60.0–71.5)	0.66 (0.46–0.94); 0.020; 0.060
rs870849 T	102 (37.2; 31.5–43.0)	115 (41.1; 35.3–46.8)	0.85 (0.61–1.20); 0.354; 0.898	115 (44.2; 38.2–50.3)	89 (34.2; 28.5–40.0)	1.52 (1.07–2.17); 0.020; 0.060

**Table 3 ijms-25-13403-t003:** Age at onset or tremor according to the genotypes.

	Age at Onset (SD); Range	*t*-Test *p*-Value	*t*-Test *p*-Value
**Genotype**		rs1922452 A/G	rs1922452 G/G
rs1922452 A/A	48.94 (17.73); 16–78	0.500	0.677
rs1922452 A/G	51.45 (18.53); 5–80		0.131
rs1922452 G/G	47.27 (20.00); 10–78		
**Genotype**		rs951818 A/C	rs951818 C/C
rs951818 A/A	46.89 (20.61); 10–78	0.097	0.651
rs951818 A/C	51.61 (18.35); 5–80		0.390
rs951818 C/C	48.65 (17.07); 16–78		
**Genotype**		rs870849 C/T	rs870849 T/T
rs870849 C/C	48.63 (20.97); 5–80	0.629	0.841
rs870849 C/T	50.00 (18.35); 12–78		0.870
rs870849 T/T	49.43 (16.77); 12–80		

## Data Availability

All data relating to the current study, intended for reasonable use, is available from J.A.G. Agúndez (University Institute of Molecular Pathology Biomarkers, University of Extremadura -UNEx ARADyAL Instituto de Salud Carlos III, Av/de la Universidad S/N, E10071 Cáceres, Spain) and F.J. Jiménez-Jiménez (Section of Neurology, Hospital del Sureste, Arganda del Rey, Madrid, Spain).

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
