# Peer review of "Association Between Common Variants in the LAG3/CD4 Genes and Risk for Essential Tremor"

_ijms, 2024, doi:10.3390/ijms252413403_

Round 1

Reviewer 1 Report

Comments and Suggestions for Authors

General aspects:

The style and grammar could be improved. Very long sentences and digressions unease reading and understanding the subject.

The hypothesis of the study is not clearly stated.

The text, including reason to study, the goal, working hypothesis and Discussion is incoherent.

This is not review as Editors marked but original paper.

Specific aspects:

Abstract:

What is the protein coded by CD4 and LAG3 genes?

Why LAG3 was chosen for this study?

What are the available gene variants?

What is the difference between CD4 rs1922452, CD4 rs951818?

In conclusion: “except for the slight decrease in risk in patients carrying the variant LAG3 rs870849C. “ when we talk about risks these are not patients yet. Also, the study result indicated only males. Please specify in conclusion.

Introduction:

-“very high prevalence” – please give numbers. Actual if possible.

-Please refrain from digressions (in parenthesis), and if important add another sentence, or if not important, skip it. Otherwise it disturbs reading flow and understanding of the subject.

-“ The higher concordance… the hig(er) positivity of family history”. Also, divide this sentence into at least three separate ones.

-“ linkage studies” -?

The hypothesis of the study is not clearly stated.

Patients and methods:

Describe clearly diagnostic criteria. Inclusion and exclusion criteria.

Since the hypothesis assumed ET and PD having related genetic risks – why exclude PD patients from the study?

Why normal thyroid function was required?

First part of Discussion is actually an introduction.

The Discussion is almost non-existing. Since authors identified that “men with ET showed a significant decrease in the frequency of LAG3 rs870849CC genotype and LAG3 rs870849C allele” – it should be discussed.

Comments on the Quality of English Language

The style and grammar could be improved. Very long sentences and digressions unease reading and understanding the subject.

Author Response

General aspects:

The style and grammar could be improved. Very long sentences and digressions unease reading and understanding the subject. OK, English style and grammar have been revised by an English native with expertise in editing scientific texts

The hypothesis of the study is not clearly stated. The hypothesis is the relation between inflammation and ET, and the attempt to search for common genes between ET and Parkinson's diseases. Some variants of the LAG3/CD4 genes related to inflammation have been associated with the risk for Parkinson's disease in some recent studies. We have added a small paragraph about studies of inflammation markers in ET, which are discussed in more detail in the discussion

The text, including reason to study, the goal, working hypothesis and Discussion is incoherent. The working hypothesis, the reason to carry out the study, and the discussion have been modified according to the specific suggestions.

This is not review as Editors marked but original paper.  Indeed, the article is original. We erroneously mared it as review by error, when uploading it in the system.

Specific aspects:

Abstract:

What is the protein coded by CD4 and LAG3 genes?  This is now specified in the introduction.

Why LAG3 was chosen for this study? Because its relation with CD4 and with inflammation. This is specified in the abstract.

What are the available gene variants? The SNVs selected were the most common for these genes, this was specified previously

What is the difference between CD4 rs1922452, CD4 rs951818?    

These are two different SNVs affecting the same gene. The rs1922452 variant consists of an A to G substitution in chromosome 12-6896194 (GRCh37), whereas the rs951818 variant consists of an C to A substitution in chromosome 12-6896055 (GRCh37). Additional information for all SNVs analyzed has been added to the Methods section.

In conclusion: “except for the slight decrease in risk in patients carrying the variant LAG3 rs870849C. “ when we talk about risks these are not patients yet. Also, the study result indicated only males. Please specify in conclusion. OK, specified

Introduction:

-“very high prevalence” – please give numbers. Actual if possible. OK, we have added these data from a recent meta-analysis of this issue (Song P, Zhang Y, Zha M, Yang Q, Ye X, Yi Q, Rudan I. The global prevalence of essential tremor, with emphasis on age and sex: A meta-analysis. J Glob Health. 2021 Apr 10;11:04028. doi:10.7189/jogh.11.04028)

-Please refrain from digressions (in parenthesis), and if important add another sentence, or if not important, skip it. Otherwise it disturbs reading flow and understanding of the subject. OK, corrected

-“ The higher concordance… the hig(er) positivity of family history”. Also, divide this sentence into at least three separate ones. OK. We have modified this sentence y 3 separate ones.

-“ linkage studies” -? Genetic linkage studiesThe hypothesis of the study is not clearly stated. T

The hypothesis is the relation between inflammation and ET, and the attempt to search for common genes between ET and Parkinson's diseases. Some variants of the LAG3/CD4 genes related to inflammation have been associated with the risk for Parkinson's disease in some recent studies. We have added a small paragraph about studies of inflammation markers in ET, which are discussed in more detail in the discussion

Patients and methods:

Describe clearly diagnostic criteria. Inclusion and exclusion criteria.

Inclusion criteria for patients with ET were:

  1. Standard diagnostic criteria for the diagnosis of definite ET, according to Deuschl, G.; Bain, P.; Brin, M. Consensus Statement of the Movement Disorder Society on Tremor. Ad Hoc Scientific Committee. Mov. Disord. 1998, 13(Suppl. 3), 2–23. https://doi.org/10.1002/mds.870131303. These were the usual criteria when the samples collection for this study began.
  2. Having at least one first-degree relative also diagnosed with ET (that is, familial ET)
  3. No family history of other neurological diseases, including PD. Therefore, patients with family history of other neurological diseases were excluded.
  4. Normal thyroid function studies, that is, patients with abnormal thyroid function studies were excluded.

Inclusion criteria for controls were:

  1. Absence of a personal or familial history of tremor. Therefore, subjects with personal or familial history of tremor.
  2. Absence of personal or familial history of PD or other movement disorders. Therefore, absence of personal or familial history of PD or other movement disorders were excluded.

These criteria were stated in the Patients and Methods section in the first version of the manuscript.

Since the hypothesis assumed ET and PD having related genetic risks – why exclude PD patients from the study?

This study focused on patients with ET. Two previous studies, mentioned in the introduction, (one of these by our group) showed association of two CD4 gene variants with PD.

  1. Guo, W.; Zhou, M.; Qiu, J.; Lin, Y.; Chen, X.; Huang, S.; Mo, M.; Liu, H.; Peng, G.; Zhu, X.; Xu, P. Association of LAG3 Genetic Variation With an Increased Risk of PD in Chinese Female Population. J. Neuroinflammation 2019, 16, 270. https://doi.org/10.1186/s12974-019-1686-z.
  2. García-Martín, E.; Pastor, P.; Gómez-Tabales, J.; Alonso-Navarro, H.; Álvarez, I.; Buongiorno, M.; Cerezo-Aris, M.O.; Aguilar, M.; Agúndez, J.A.G.; Jiménez-Jiménez, F.J. Association Between LAG3/CD4 Gene Variants and Risk for Parkinson’s Disease. Eur. J. Clin. Invest. 2022, 00, e13847. https://doi.org/10.1111/eci.13847.

Why normal thyroid function was required?

Normal thyroid function was required to avoid confounders in the diagnosis, because thyroid dysfunction, specially hyperthyroidism, can induce tremor.

First part of Discussion is actually an introduction. The Discussion is almost non-existing. Since authors identified that “men with ET showed a significant decrease in the frequency of LAG3 rs870849CC genotype and LAG3 rs870849C allele” – it should be discussed.

We have added a small paragraph about studies of inflammation markers in ET, although we discussed them in more detail in the discussion section. We also clarified that, after applying corrections for multiple comparisons, this significance is marginal.

The style and grammar could be improved. Very long sentences and digressions unease reading and understanding the subject. OK, English style and grammar have been revised by an English native with expertise in editing scientific texts

Reviewer 2 Report

Comments and Suggestions for Authors

General Comments: This manuscript investigates potential associations between LAG3/CD4 genetic variants and Essential Tremor (ET) risk in a Spanish population. While the study is generally well-conducted from a technical standpoint, several major concerns need to be addressed:

Major Concerns:

  1. Result Interpretation and Claims 

The authors' conclusion of "no association" between LAG3/CD4 variants and ET appears inconsistent within the paper. In the abstract and discussion, they claim no association, yet they also emphasize a "decreased risk for ET in carriers of the LAG3 rs870849 C/C genotype and the LAG3 rs870849C allelic variant exclusively in males." The authors need to be consistent in their interpretation and presentation of results. If they conclude there is no association, they should avoid highlighting marginal findings that don't survive multiple testing correction. If they believe there are suggestive associations, they should present these as preliminary findings requiring validation.

  1. Statistical Significance and Interpretation

None of the reported associations reach statistical significance after FDR correction (all adjusted p-values > 0.05) and the findings are far from reaching genome-wide significance levels. The authors should be more cautious in interpreting and presenting these marginally significant results. The term "association" should be used more carefully when describing findings that don't meet statistical thresholds

  1. Research Rationale

The primary justification for studying these variants (previous association with PD) is insufficient. Genetic overlap between neurological disorders is common and often non-specific. A stronger biological rationale for why these specific variants might influence ET pathogenesis is needed. The authors should consider and discuss other neurological conditions showing LAG3/CD4 associations, or indicate PD is the only neurological condition if that’s the case. 

  1. Mechanistic Understanding

The paper lacks sufficient mechanistic explanation linking LAG3 to gender specific ET pathogenesis. There’s no clear pathway connecting LAG3 function to tremor generation. The gender-specific effects observed lack biological explanation. Functional predictions for the variants studied are not included.

Suggestions for Improvement:

  1. Data Presentation

Author should add graphical representations for their key findings, like using Forest plots showing odds ratios and 95% CI for all variants, as well as a Comparative analysis of effect sizes between PD and ET for these variants to help understand relative impact of these variants in different disorders.

Author can add age distribution plots comparing cases (current age and age of onset) and controls. Author has acknowledged the potential issue for age mistach in their study, this visualization would strengthen the paper's transparency regarding the age-matching limitation and help readers better assess its potential impact on the findings.

2. Statistical Analysis

The statistical methodology requires several important improvements. Power calculations for subgroup analyses, particularly for the gender-specific findings, should be included to help readers evaluate the reliability of these results. The multiple testing correction methodology needs clearer explanation, including justification for the chosen FDR approach and consideration of the total number of tests performed. Additionally, while the paper mentions Hardy-Weinberg equilibrium testing was performed, these results should be presented in tables for completeness and transparency.

3. Mechanistic Framework 

The mechanistic underpinning of this study needs substantial strengthening. The authors should provide a more robust biological rationale for investigating these specific variants in ET, including functional predictions for how these variants might affect LAG3 protein function or expression. The paper would benefit from a detailed discussion of LAG3 expression patterns in tremor-relevant circuits and how alterations in these circuits might contribute to ET pathogenesis. A clear mechanistic hypothesis linking LAG3 function to tremor generation should be presented, preferably supported by existing literature or preliminary functional data.

4. Discussion: 

The authors should provide more specific suggestions for future validation studies, including recommended sample sizes and potential functional studies to explore the observed gender-specific effects.

Minor Points: To enhance the overall clarity and completeness of the manuscript, several additions are recommended. A flow diagram illustrating patient selection criteria and process would help readers understand the study population better. More detailed demographic information, including clinical characteristics of the ET cases, should be provided. Finally, the authors should consider adding linkage disequilibrium analysis between the studied variants to better understand their relationships and potential combined effects.

While this study presents interesting preliminary data, substantial revisions are needed to address the statistical, mechanistic, and methodological concerns before it can make a meaningful contribution to the field. The authors should consider presenting this as a hypothesis-generating study rather than a confirmatory analysis.

Author Response

General Comments: This manuscript investigates potential associations between LAG3/CD4 genetic variants and Essential Tremor (ET) risk in a Spanish population. While the study is generally well-conducted from a technical standpoint, several major concerns need to be addressed:

Major Concerns:

  1. Result Interpretation and Claims 

The authors' conclusion of "no association" between LAG3/CD4 variants and ET appears inconsistent within the paper. In the abstract and discussion, they claim no association, yet they also emphasize a "decreased risk for ET in carriers of the LAG3 rs870849 C/C genotype and the LAG3 rs870849C allelic variant exclusively in males." The authors need to be consistent in their interpretation and presentation of results. If they conclude there is no association, they should avoid highlighting marginal findings that don't survive multiple testing correction. If they believe there are suggestive associations, they should present these as preliminary findings requiring validation.

  1. We clarified that, after applying corrections for multiple comparisons, this significance of this finding was marginal (Pc = 0.06). We added that these data might be considered as preliminary. The need of replication in future studies was commented in the first version. The abstract has been modified accordingly.
  2. Statistical Significance and Interpretation

None of the reported associations reach statistical significance after FDR correction (all adjusted p-values > 0.05) and the findings are far from reaching genome-wide significance levels. The authors should be more cautious in interpreting and presenting these marginally significant results. The term "association" should be used more carefully when describing findings that don't meet statistical thresholds

  1. We clarified that, after applying corrections for multiple comparisons, this significance of this finding was marginal (Pc = 0.06)
  2. Research Rationale

The primary justification for studying these variants (previous association with PD) is insufficient. Genetic overlap between neurological disorders is common and often non-specific. A stronger biological rationale for why these specific variants might influence ET pathogenesis is needed. The authors should consider and discuss other neurological conditions showing LAG3/CD4 associations, or indicate PD is the only neurological condition if that’s the case. 

  1. The hypothesis is the relation between inflammation and ET, and the attempt to search for common genes between ET and Parkinson's diseases. Some variants of the LAG3/CD4 genes related to inflammation have been associated with the risk for Parkinson's disease in some recent studies. We have added a small paragraph about studies of inflammation markers in ET, which are discussed in more detail in the discussion.

We commented in the first version of the discussion on other neurological and non-neurological conditions showing LAG/CD4 associations, including multiple sclerosis, PD, disease progression and mortality of sepsis, and the severity of primary immune thrombocytopenia (rs870849 T>C) .

  1. Mechanistic Understanding

The paper lacks sufficient mechanistic explanation linking LAG3 to gender specific ET pathogenesis. There’s no clear pathway connecting LAG3 function to tremor generation. The gender-specific effects observed lack biological explanation. Functional predictions for the variants studied are not included.   

  1. We have added the following paragraph in the introduction: “CD4 protein is expressed in the brain of adult rats, including in neurons from the cerebellar cortex (both in granule cells and in Purkinje cells) and thalamus (implicated in the generation of tremor), and to a lesser extent in the striatum and substantia nigra compacta [23]. In the human brain, CD4 is highly expressed in the thalamus and basal ganglia and has some degree of expression in the cerebellum as well [24]. The presence or absence of LAG3 protein in the brain is under debate, with some publications indicating its presence in certain brain regions and other that found lack of evidence of its presence in neuronal cell lines [25].

 Suggestions for Improvement:

  1. Data Presentation

Author should add graphical representations for their key findings, like using Forest plots showing odds ratios and 95% CI for all variants, as well as a Comparative analysis of effect sizes between PD and ET for these variants to help understand relative impact of these variants in different disorders.  

Done, two Forest plots have been added. The first one shows the main findings for ET patients, as a single group and classified by sex. The second figure compares the data obtained in TE and PD patients, classified by gene variants.

Author can add age distribution plots comparing cases (current age and age of onset) and controls. Author has acknowledged the potential issue for age mistach in their study, this visualization would strengthen the paper's transparency regarding the age-matching limitation and help readers better assess its potential impact on the findings.  Done, we included a violin plot comparing the age at onset, the current age of patients and the age of controls.

  1. Statistical Analysis

The statistical methodology requires several important improvements. Power calculations for subgroup analyses, particularly for the gender-specific findings, should be included to help readers evaluate the reliability of these results.

The requested power calculations were included in the revised manuscript.

The multiple testing correction methodology needs clearer explanation, including justification for the chosen FDR approach and consideration of the total number of tests performed.

Multiple testing was assessed by using False Discovery Rate, which is adequate for the control of type I error. This procedure controls the proportion of wrongly rejected null hypotheses amongst those that are rejected (instead of amongst all). This procedure is more sensible when there is a high number of comparisons as in this study: three genotypes x three groups (overall patients, men and women) x five groups according to tremor localization. This is now stated in the revised manuscript.

Additionally, while the paper mentions Hardy-Weinberg equilibrium testing was performed, these results should be presented in tables for completeness and transparency.

Done. The results for Hardy Weinberg’s equilibrium were included in the revised version of the manuscript.

  1. Mechanistic Framework 

The mechanistic underpinning of this study needs substantial strengthening. The authors should provide a more robust biological rationale for investigating these specific variants in ET, including functional predictions for how these variants might affect LAG3 protein function or expression. The paper would benefit from a detailed discussion of LAG3 expression patterns in tremor-relevant circuits and how alterations in these circuits might contribute to ET pathogenesis. A clear mechanistic hypothesis linking LAG3 function to tremor generation should be presented, preferably supported by existing literature or preliminary functional data. 

  1. We have added the following paragraph in the introduction: “CD4 protein is expressed in the brain of adult rats, including in neurons from the cerebellar cortex (both in granule cells and in Purkinje cells) and thalamus (implicated in the generation of tremor), and to a lesser extent in the striatum and substantia nigra compacta [23]. In the human brain, CD4 is highly expressed in the thalamus and basal ganglia and has some degree of expression in the cerebellum as well [24]. The presence or absence of LAG3 protein in the brain is under debate, with some publications indicating its presence in certain brain regions and other that found lack of evidence of its presence in neuronal cell lines [25].
  2. Discussion: 

The authors should provide more specific suggestions for future validation studies, including recommended sample sizes and potential functional studies to explore the observed gender-specific effects. OK, added at the end of the discussion.

Minor Points: To enhance the overall clarity and completeness of the manuscript, several additions are recommended.

A flow diagram illustrating patient selection criteria and process would help readers understand the study population better.

More detailed demographic information, including clinical characteristics of the ET cases, should be provided.

Done, clinical characteristics (tremor location) have been included in the revised manuscript and genotype comparison in subgroups of patients according such characteristics were carried out.

Finally, the authors should consider adding linkage disequilibrium analysis between the studied variants to better understand their relationships and potential combined effects.

Results of LD analyses were added to the revised version of the manuscript.

Round 2

Reviewer 2 Report

Comments and Suggestions for Authors

All my comments has been addressed. Only have 1 more suggestions about title of the paper: consider change it to "Investigation of LAG3/CD4 Gene Variants in Essential Tremor: A Case-Control Study", current one might be misleading to readers because it suggests there is an association when their findings actually show no significant association after proper statistical correction.

Author Response

Only have 1 more suggestions about title of the paper: consider change it to "Investigation of LAG3/CD4 Gene Variants in Essential Tremor: A Case-Control Study", current one might be misleading to readers because it suggests there is an association when their findings actually show no significant association after proper statistical correction.

OK, TITLE CHANGED AS INCICATED